# Oncogenic human papillomavirus and anal microbiota in men who have sex with men and are living with HIV in Northern Taiwan

Shu-Hsing Cheng[1,2‡], Yu-Chen Yang[3‡], Cheng-Pin Chen[1,4], Hui-Ting Hsieh[1], Yi-Chun Lin[1,5], Chien-Yu Cheng[1,6], Kuo-Sheng Liao[7], Fang-Yeh Chu[8,9,10‡*], Yun-Ru Liu[3‡*]

1 Department of Infectious Diseases, Taoyuan General Hospital, Ministry of Health and Welfare, Taoyuan, Taiwan, 2 School of Public Health, Taipei Medical University, Taipei, Taiwan, 3 Joint Biobank, Office of Human Research, Taipei Medical University, Taipei, Taiwan, 4 Institute of Clinical Medicine, National Yang Ming Chiao Tung University, Taipei, Taiwan, 5 Graduate Institute of Clinical Medicine, College of Medicine, Taipei Medical University, Taipei, Taiwan, 6 Institute of Public Health, National Yang Ming Chiao Tung University, Taipei, Taiwan, 7 Department of Pathology, Taoyuan General Hospital, Ministry of Health and Welfare, Taoyuan, Taiwan, 8 Department of Clinical Pathology, Far Eastern Memorial Hospital, New Taipei City, Taiwan, 9 Graduate School of Biotechnology and Bioengineering, Yuan Ze University, Taoyuan City, Taiwan, 10 School of Medical Laboratory Science and Biotechnology, Taipei Medical University, Taipei, Taiwan

‡ SHC and YCY contributed equally as 1st authors. FYC and YRL contributed equally as corresponding authors.
* jacpha@mail.femh.org.tw (FYC); d90444002@tmu.edu.tw (YRL)

**Data Availability Statement:** The data that support the findings of this study are available in the Figshare database, doi: 10.6084/m9.figshare.25375087.

## Abstract

Few studies have demonstrated the interplay between human immunodeficiency virus (HIV), anal human papillomavirus (HPV), and anal microbiota, especially in persons living with HIV who are men who have sex with men. We, therefore, explored these interrelationships in a cohort of persons living with HIV, mainly comprising men who have sex with men. HPV genotyping using a commercial genotyping kit and ThinPrep cytology interpreted by Bethesda systems was performed on samples from 291 patients. Samples were characterized by high-throughput sequencing of dual-index barcoded 16s rRNA (V3–4). Bacterial diversity was diminished in individuals living with HIV with CD4+ T cells <500 cells/μL and anal cytology yielding atypical squamous cells of undetermined significance or higher grades (ASCUS+) with detectable HPV 16/18 compared with those with CD4+ T cells ≥500 cells/μL with ASCUS+ and HPV 16/18 and those with normal anal cytology or inflammation without HPV 16/18. Enterobacteriaceae, *Ruminococcus*, and Bacilli were significantly abundant in persons living with HIV with CD4+ T cells <500 cells/μL with ASCUS+ and HPV 16/18. Bacterial diversity, composition, and homogeneity of dispersion were different in individuals living with HIV with low CD4+ T cells with ASCUS+ and HPV 16/18, and understanding the interaction among immunocompromised hosts, oncogenic HPVs, and microbiota is essential, and the contribution of these factors to anal precancerous lesions needs more in-depth exploration.

**Funding:** SHC received funding for this work from Taoyuan General Hospital [grant numbers: PTH10715, PTH10805, PTH10915, and PTH11009]. https://www.tygh.mohw.gov.tw/ The sponsors or funders did not play any role in the study design, data collection and analysis, decision to publish, or preparation of the manuscript.

**Competing interests:** The authors have declared that no competing interests exist.

## Introduction

Human papillomavirus (HPV) is associated with cancer, including cervical, head and neck, and genital cancers, and anal squamous cell carcinoma [1, 2]. Certain populations have been noted to have a higher risk of developing anal cancer, namely persons living with human immunodeficiency virus (HIV) and men who have sex with men (MSM) [3]. Persistent oncogenic HPV infections were noted in persons living with HIV (PLWH), especially with immunocompromised status (CD4 + T cell counts less than 200 µL/mL) [4]. Furthermore, the incidence of HPV-related cancer continues to increase in the era of highly active antiretroviral therapy (HAART) in PLWH [2, 5].

Recently, there have been many studies on the symbiotic relationship between the host and indigenous microbiota in establishing viral infections and the progression of virus-related diseases. Changes in commensal microbiota may affect the prognosis of influenza, norovirus, and COVID-19 [6–8]. Dysbiosis has been implicated in carcinogenic pathways, including the progression from chronic hepatitis B or C to hepatocellular carcinoma [9–11], and the association of HPV with cervical carcinogenesis in women [12, 13]. Intestinal dysbiosis and chronic immune activation interact in PLWH [14, 15]. *Peptostreptococcus* and Fusobacteria have been implicated in the pathogenesis of colorectal cancers [16, 17] and are associated with poor prognosis [18].

There are fruitful studies on the interaction between the microbiota, HPV acquisition and persistence, and the development of cervical cancer [13, 19, 20]. Women with cervical intraepithelial neoplasia exhibit a microbiota profile with increased bacterial diversity and a reduced abundance of *Lactobacillus* spp. Ravel et al. [21] described bacterial vaginosis-associated bacteria (community state type IV) as a heterogeneous group characterized by the depletion of *Lactobacillus* spp. and the presence of anaerobic species like *Gardnerella*, *Megasphaera*, *Sneathia*, and *Prevotella*. Lactobacilli provide broad-spectrum protection in the vagina [22]. An imbalance, such as a depletion of *Lactobacillus* and an increase in other microbiota, can lead to histological alterations in the vaginal mucosa and cervical epithelium [23, 24]. A previous meta-analysis showed that bacterial vaginosis is associated with higher rates of HPV infection (odds ratio 1.43, 95% CI 1.11–1.84) [25], suggesting that a *Lactobacillus*-depleted microbiome may contribute to HPV persistence in the cervix. In addition, bacterial vaginosis may involve reactive oxygen species that cause DNA damage, facilitating HPV integration and malignant transformation [23]. Women living with HIV (PLWH) were found to have cervical lesions with increased bacterial diversity and reduced *Lactobacillus* spp. [26, 27], similar to that in women without HIV [13, 19]. However, studies on anal lesions, HPVs, and anal mucosa microbiota are limited [20, 28, 29]. In this study, we explored the interrelationships between HIV, anal HPV, anal precancerous lesions, and anal microbiota in a PLWH cohort comprising MSM (MSM-LWH).

## Materials and methods

### Study participants

Between Jun 1, 2018, and Dec 31, 2021, MSM-LWH visited one of the outpatient clinics of Taoyuan General Hospital, Taiwan, voluntarily enrolled in this study. Taoyuan General Hospital is a 1,020-bed referral hospital in northern Taiwan serving 2.3 million people, more than 4,300 of whom live with HIV [30]. After providing written informed consent, the participants completed a self-administered questionnaire (S1 File) that addressed their education level; marital status; sexual behavior (heterosexuality or homosexuality, lifetime number of sexual partners, number of new sexual partners within the previous 6 months, frequency of receptive anal sex (always, often, occasionally, seldom, or never), frequency of condom use during anal

sex (always, often, occasionally, seldom, or never), frequency of chemsex (always, often, occasionally, seldom, or never); self-reported sexually transmitted infections (STIs) within the previous 6 months (syphilis, gonorrhea, chlamydial urethritis, condyloma acuminata, amebic colitis/liver abscess, or other clinical diagnoses of STIs), and history of HPV vaccination. Saline-wetted Dacron swabs (Amplicor STD Swab Specimen Collection and Transport Set; Roche Molecular Systems, Branchburg, NJ, USA) were inserted approximately 5 cm above the anal verge. Rectal swabs were immediately rinsed in vials of PreservCyt solution (Hologic Inc., Marlborough, MA, USA) and Omnigene collection tubes (OMR-130; DNA Genotek Inc., Stittsville, ON, Canada).

This study was approved by the Institutional Review Board of Taoyuan General Hospital (IRB#: TYGH106042, TYGH10743 and TYGH108050).

## Anal Pap smears

Anal cytology samples were prepared using thin-preparation Pap smears (ThinPrep; Hologic Inc., Marlborough, MA, USA) and sent to a certified laboratory for blinded interpretation by two cytopathology technicians and two pathologists. The results were classified according to the 2001 Bethesda System [31]. We considered the following as anal cellular dysplasia: atypical squamous cells of undetermined significance (ASCUS), low-grade squamous intraepithelial lesions (LSILs), high-grade squamous intraepithelial lesions (HSILs), and atypical squamous cells that could not exclude HSIL (ASC-H). The cells were preserved in PreservCyt solution and stored at −70°C for DNA testing.

## Human papillomavirus detection

HPV detection was performed using polymerase chain reaction (PCR) (Cobas HPV; Roche Molecular Systems, Branchburg, NJ, USA). Fourteen types of HPV have been detected, including oncogenic types 16 and 18, as well as pooled results of types 31, 33, 35, 39, 45, 51, 52, 56, 58, 59, 66, and 68.

## Targeted amplicon library preparation and sequencing

16S rRNA was purified from anal samples by QIAamp Fast DNA Stool Mini Kit (QIAGEN, Germany). The 16S rRNA gene amplification and library construction were performed according to Illumina's recommended protocols [32]. Briefly, the universal primers 341F (5′–CCTACGGGNGGCWGCAG–3′) and 805R (5′–GACTACHVGGGTATCTAATCC–3′) containing Illumina overhang adapter sequences in the forward (5′–TCGTCGGCAGCGTCAGATGTGTAT AAGAGACAG–3′) and reverse (5′–GTCTCGTGGGCTCGGAGATGTGTATAAGAGACAG–3′) primers were used to amplify the V3–V4 region of bacterial 16S rRNA using a limited cycle PCR. A Nextera XT Index kit (Illumina Inc., San Diego, CA, USA) was used to add Illumina sequencing adapters and dual-index barcodes to the amplicon targets. Quantification and quality of the sequencing libraries were checked using a QSep100 Analyzer (BiOptic Inc., New Taipei City, Taiwan). Finally, the libraries were normalized and pooled in an equimolar ratio by Nextera XT DNA Library Preparation Kit (Illumina) and sequenced using an Illumina Miseq instrument MiSeq Reagent Kit V3 (600 cycle) (Illumina Inc., San Diego, CA, USA).

## 16S rRNA gene sequence analysis

Universal primer sequences and low-quality reads were trimmed using Cutadapt (v1.18) [33] and then processed and analyzed using the DADA2/phyloseq workflow in the R environment. Briefly, filtering, trimming, dereplication, and denoising of forward and reverse reads were

performed using DADA2 (v1.10.0) [34]. Batch effects were compensated using the DESeq2 package in R, with a maximum of 96 indexes per run [35]. Processed reads were merged, and chimeras were subsequently removed from the cleaned full-length amplicons. Taxonomic assignment of the inferred amplicon sequence variants was performed using the SILVA reference database (v138) [36] with a minimum bootstrap confidence of 80. Multiple sequence alignments of ASVs were performed using DECIPHER (v2.8.1) [37], and a phylogenetic tree was constructed using RaxML (v8.2.11) [38]. The frequency table, taxonomy, and phylogenetic tree information were used to create a phyloseq object for downstream bacterial community analyses using phyloseq (v1.30.0) [39]. Alpha diversity was measured, which indicated the richness and evenness of bacterial taxa within a community, calculated by using the estimate_richness function from the phyloseq package with observed, chao index, Shannon, and Simpson methods; Beta diversity, which reflects the extent of variation in microbial community composition between the two groups, using Principal Co-ordinates Analysis (PcoA) on unweighted Unique Fraction (UniFrac) and weighted UniFrac, PcoA on variance-adjusted weighted UniFrac; and PcoA on Generalized Unique Fraction (GUniFrac) with alpha 0.5. UniFrac distances were calculated using the GUniFrac package (v1.1) to assess the community dissimilarity between groups [40]. Non-metric Multidimensional Scaling (NMDs) on Bray–Curtis distance (VST), Adonis tests (to examine whether composition between groups was similar), and the Betadisper test (to test the homogeneity of dispersion between groups, which is an assumption of Adonis) was calculated using DESeq2 package (V1.26.0) [33]. Microbiota enrichment analysis was conducted using the Linear Discriminant Analysis (LDA) effect size (LefSe) method, applying the Wilcoxon–Mann–Whitney test ($\alpha = 0.05$) and logarithmic LDA score > 2 [41]. The results were visualized as a cladogram using GraPhlAn [42].

## Statistical analyses

Demographic data are presented as mean ± standard deviation (SD) for continuous variables, median (25th, 75th percentile), and percentiles for discrete variables. The distribution of the cytology grading was calculated, and the HPV genotype results were analyzed. Chi-square tests were used to compare categorical variables, while Wilcoxon–Mann–Whitney tests were used to compare continuous variables. Anal samples were divided into six groups; group 1: samples from PLWH whose CD4+ T cell counts were < 500 cells/μL with anal cytology yielding ASCUS or higher grades (ASCUS+) and detectable HPV 16/18; group 2: samples from PLWH whose CD4+ T cell counts were ≥ 500 cells/μL with anal cytology yielding ASCUS+ with detectable HPV 16/18; group 3: samples from PLWH who had anal cytology yielding ASCUS + without HPV 16/18 detection, group 4: samples from PLWH whose CD4+ T cell counts were < 500 cells/μL and who had normal anal cytology or inflammation with detectable HPV 16/18; group 5: samples from PLWH whose CD4+ T cell counts ≥ 500 cells/μL and who had normal anal cytology or inflammation with detectable HPV 16/18; and group 6: samples from PLWH who had normal anal cytology or inflammation without HPV 16/18 detection. Each group (groups 2, 3, 4, 5, and 6) was compared to group 1. P < 0.05 was considered statistically significant. All statistical analyses were conducted using SAS 9.3 software (SAS Institute, Inc., Cary, NC, USA).

## Results

In total, 338 anal samples were collected. However, 10 (2.9%) failed cytology interpretation, 28 (8.3%) failed HPV detection tests, and 9 (2.7%) were from heterosexual PLWH. Finally, 291 (86.1%) samples collected from MSM-LWH were analyzed (Table 1).

**Table 1. Demographic characteristics, behavior risks, and immunologic factors for the study cohort of men who have sex with men and live with HIV in northern Taiwan.**

| Characteristics | Patients N = 291 | Percentage, (SD), or (25, 75$^{th}$ quartile) |
|---|---|---|
| Age, mean (SD) | 39.26 | (10.06) |
| Median (25$^{th}$, 75$^{th}$ quartile) | 37 | (32, 47) |
| Years in school | | |
| 6 to 9 | 14 | 4.8% |
| >9 to 12 | 80 | 27.5% |
| >12 | 197 | 67.7% |
| Marriage | | |
| Single | 271 | 93.1% |
| Others | 20 | 6.9% |
| HIV years, mean (SD) | 8.55 | (5.55) |
| Median (25$^{th}$, 75$^{th}$ quartile) | 8 | (4, 11) |
| Smoking | 115 | 39.5% |
| Drinking | 116 | 39.9% |
| Betelnut | 9 | 3.1% |
| Lifetime sexual partner, mean (SD) | 21.62 | (50.74) |
| Median (25$^{th}$, 75$^{th}$ quartile) | 10 | (5;20) |
| Sexual partner in 1/2 y, mean (SD) | 2.17 | (4.33) |
| Median (25$^{th}$, 75$^{th}$ quartile) | 1 | (1,2) |
| Condom use during anal sex | | |
| Every time, often | 156 | 53.6% |
| Occasional | 114 | 39.2% |
| Rare, never | 21 | 7.2% |
| Met sexual partners on the web | 196 | 67.4% |
| STDs in 1/2 y | 85 | 29.2% |
| Circumcision | 52 | 17.9% |
| HPV vaccination | 62 | 21.3% |
| Substance use in 1/2 y | 42 | 14.4% |
| Ever participated in chemsex | 95 | 32.6% |
| On cART | 291 | 100% |
| Current CD4+ T cell count, mean (SD) | 607.62 | (267.11) |
| Median (25$^{th}$, 75$^{th}$ quartile) | 598.5 | (437.75;759.75) |
| Undetectable viral load (VL<50) | 267 | 91.8% |

Abbreviations: cART: combinational antiretroviral therapy; HIV: human immunodeficiency virus; HPV: human papillomavirus; SD: standard deviation; STD: sexually transmitted diseases.

Among the anal samples from the 291 participants, 199 (68.4%) had detectable oncogenic HPVs, 92 (31.6%) did not, 68 (23.4%) were positive for HPV type 16/18, 36 (12.4%) yielded HPV 16, 38 (13.1%) yielded HPV 18, and 190 (65.3%) yielded pooled results for 12 other oncogenic HPVs (Table 2).

Among the anal samples from the 291 participants, 133 (45.7%) yielded cytology interpretations with ASCUS+, including 70 (24.1%) ASCUS, 49 (16.8%) LSIL, 2(0.7%) HSIL, and 12 (4.1%) ASC-H, and 158 (52.3%) with normal anal cytology or inflammation. The anal samples (n = 291) were divided into six groups according to HPV16/18 detection, cytology results, and participants' CD4+ T cell counts (Fig 1).

**Table 2. Oncogenic HPV distribution among 291 participants who are MSM-LWH.**

| | Total N = 291(100%) | Normal or inflammation N = 158 (100%) | ASCUS+ N = 133(100%) | P value |
|---|---|---|---|---|
| Oncogenic HPV detection | 199 (68.4%) | 89 (56.3%) | 110 (82.7%) | <0.00001 |
| HPV16 | 36 (12.4%) | 12 (7.6%) | 24 (18.0%) | 0.007 |
| HPV18 | 38 (13.1%) | 11 (6.9%) | 27 (20.3%) | 0.0007 |
| HPV16 and/or 18 | 68 (23.3%) | 22 (13.9%) | 46 (34.6%) | 0.00003 |
| Pooled 12 other oncogenic HPVs* | 190 (65.3%) | 85 (53.8%) | 105 (78.9%) | <0.00001 |

* HPV types 31, 33, 35, 39, 45, 51, 52, 56, 58, 59, 66, and 68.

Abbreviations: ASCUS+: included atypical squamous cells of undetermined significance, low-grade squamous intraepithelial lesions, high-grade squamous intraepithelial lesions (HSIL), and atypical squamous cells that could not exclude HSIL; HIV: human immunodeficiency virus; HPV: human papillomavirus, MSM: men who have sex with men; MSM-LWH: MSM who are living with HIV.

We did not find a clear difference in alpha diversity metrics between samples from group 1 and samples from other groups (observed p = 0.033 for group 1 vs. group 2; p = 0.022 for group 1 vs. group 6; Chao1 p = 0.053 for group 1 vs. group 6; all by Wilcoxon–Mann–Whitney tests, Fig 2).

We also examined beta diversity. We found differences at this level between samples from group 1 and those from the other groups (S1 Table, Fig 3).

We used the LefSe discovery tool to identify the predictors of abnormal anal cytology. This method addresses the issue of identifying microorganisms that consistently explain the differences between ≥2 microbial communities. When comparing groups 1 and 2, LefSe showed abundant Enterobacterales, Enterobacteriaceae, and *Mycoplasma* in group 1 and abundant *Finegoldia* in group 2 (Fig 4A). When comparing groups 1 and 3, LefSe showed abundant Enterobacterales and Enterobacteriaceae in group 1, and abundant *Alloprevotella* and Erysipelatoclostridiaceae in group 3 (Fig 4B). When comparing groups 1 and 4, LefSe showed

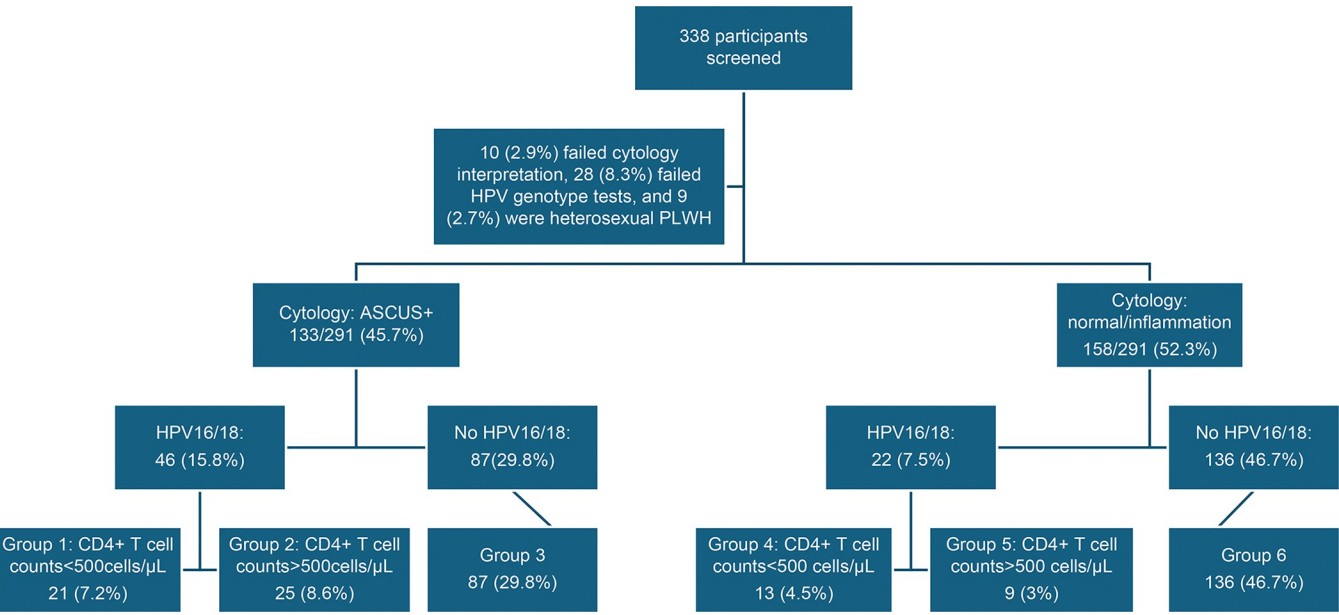

**Fig 1. Grouping of anal samples according to cytology, HPV 16/18, and CD4+ T cell counts.** ASCUS, atypical squamous cells of undetermined significance; Gr, group; HPV, human papillomavirus; MSM-LWH, men who have sex with men and live with HIV.

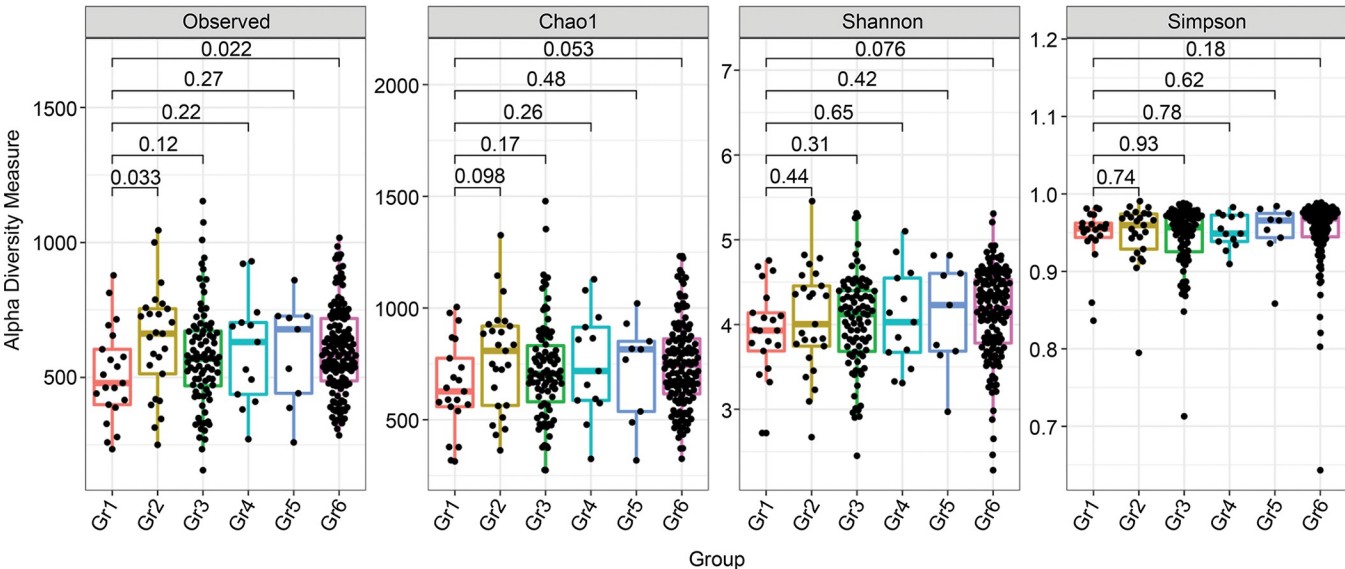

**Fig 2. Alpha diversity comparing group 1 and groups 2, 3, 4, 5, and 6.** Observed p = 0.033 for group 1 vs. group 2, p = 0.022 for group 1 vs. group 6; Chao1 p = 0.053 for group 1 vs. group 6, all by Wilcoxon–Mann–Whitney tests. Note: group 1: PLWH whose CD4+ T cell counts were < 500 cells/μL with anal cytology yielding ASCUS+ and detectable HPV 16/18 (n = 21); group 2: PLWH whose CD4+ T cell counts were ≥ 500 cells/μL with anal cytology yielding ASCUS+ with detectable HPV 16/18 (n = 25); group 3: PLWH who had anal cytology yielding ASCUS+ without HPV 16/18 detection (n = 87), group 4: PLWH whose CD4+ T cell counts were < 500 cells/μL and who had normal anal cytology or inflammation with detectable HPV 16/18 (n = 13); group 5: PLWH whose CD4+ T cell counts were ≥ 500 cells/μL and who had normal anal cytology or inflammation with detectable HPV 16/18 (n = 9); and group 6: PLWH who had normal anal cytology or inflammation without HPV 16/18 detection (n = 136). ASCUS, atypical squamous cells of undetermined significance; Gr, group; HPV, human papillomavirus; PLWH, persons living with HIV.

abundant *Ruminococcus* and *Sutterella* in group 1 and abundant Porphyromonadaceae, *Porphyromonas*, and *Fenollaria* in group 4 (Fig 4C). When comparing groups 1 and 5, LefSe did not show significant findings. When comparing groups 1 and 6, LefSe showed abundant Enterobacterales, Enterobacteriaceae, and Bacilli in group 1, and abundant Bacteroidota, Bacteroidia, Bacteroidales, and Clostridia in group 6 (Fig 4D).

## Discussion

To the best of our knowledge, this is the first research focusing on the presentations of anal microbiota among Taiwanese MSM-LWH with different stages of HIV-related diseases and levels of cellular immunity. Depletion of certain *Lactobacillus* and an increase in the richness of microbiota may contribute to HPV persistence [25], and subsequent histological alterations of the vaginal mucosa [21] and the cervical epithelium (squamous intraepithelial neoplasm) [23, 24]. This biology could be also applied to the anal mucosal environment of MSM-LWH [43]. A small study of 42 MSM-LWH revealed that *Peptostreptococcus*, *Campylobacter*, and *Gardnerella* were associated with anal precancerous lesions [28]. Another study involving 113 Nigerian MSM found that a higher proportion of MSM with prevalent HPV16 was associated with a cluster enriched in Sneathia from the family Fusobacteriaceae [29]. However, there were only six PLWH treated with HAART with viral suppression in the study by Nowak et al. [29]. Another study showed that the Ruminococcaceae NK4A214 group, *Alloprevotella* genus, *Prevotella melanonigenica*, and Ruminococcaceae UCG-014 were the most predictive of biopsy-proven HSIL [44].

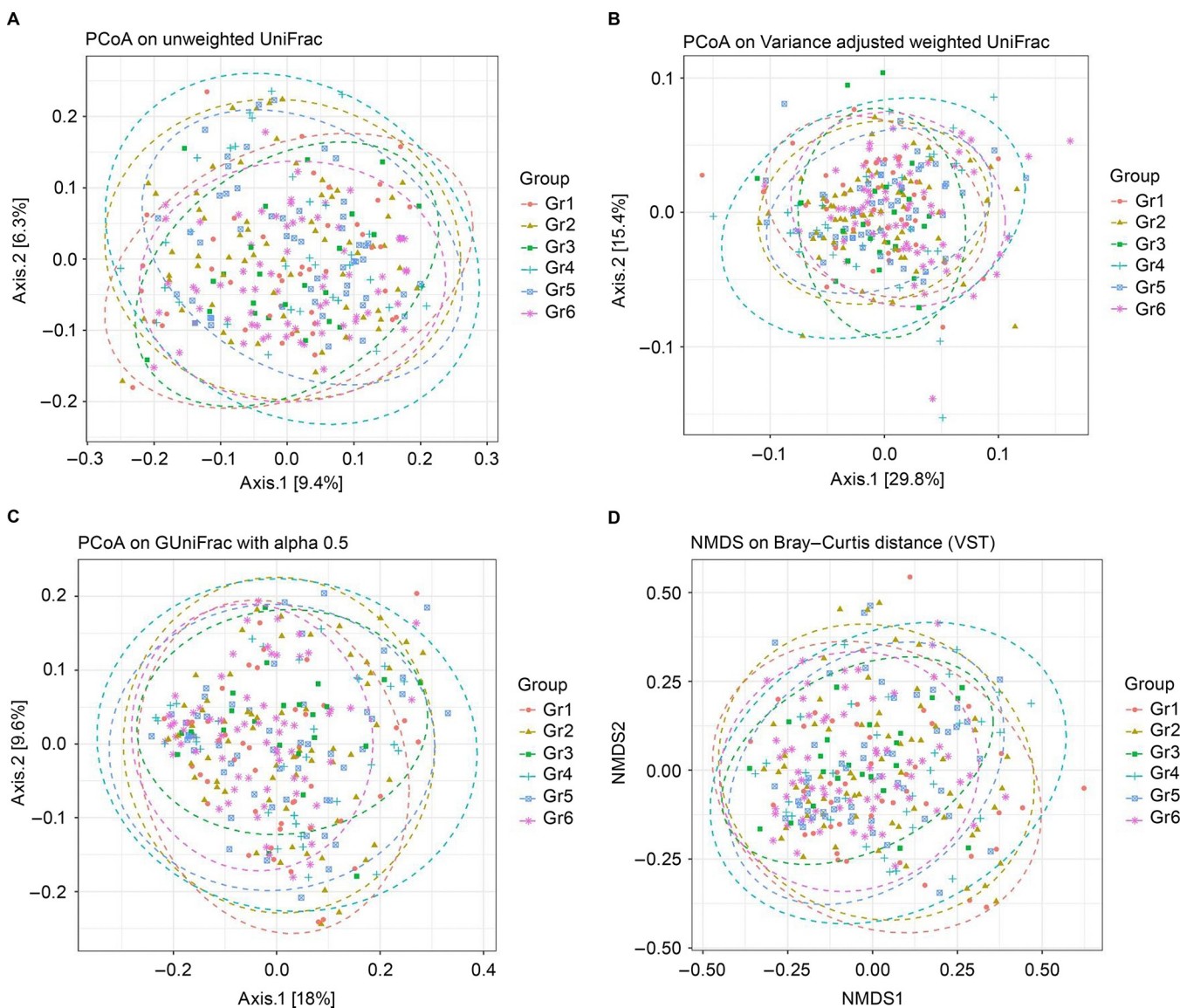

**Fig 3. Beta diversity comparing group 1 and groups 2, 3, 4, 5, and 6.** 3a: PcoA on unweighted UniFrac: Betadisper test p = 0.038 (group 1 vs. group 5), Adonis p = 0.014 (group 1 vs. group 6); 3b: no significance; 3b: PcoA on variance adjusted weighted UniFrac: Adonis p = 0.017 (group 1 vs. group 6); 3c: PcoA on GuniFrac with alpha 0.5: Adonis p = 0.030 (group 1 vs. group 6); and 3d: NMDs on Bray–Curtis distance: Adonis test p = 0.0639 (group 1 vs. group 2); Adonis test 0.0149 (group 1 vs. group 3); Adonis tests p = 0.0006 (group 1 vs. group 6). Note: group 1: PLWH whose CD4+ T cell counts were < 500 cells/μL with anal cytology yielding ASCUS+ and detectable HPV 16/18 (n = 21); group 2: PLWH whose CD4+ T cell counts were ≥ 500 cells/μL with anal cytology yielding ASCUS+ with detectable HPV 16/18 (n = 25); group 3: PLWH who had anal cytology yielding ASCUS+ without HPV 16/18 detection (n = 87), group 4: PLWH whose CD4+ T cell counts were < 500 cells/μL and who had normal anal cytology or inflammation with detectable HPV 16/18 (n = 13); group 5: PLWH whose CD4+ T cell counts were ≥ 500 cells/μL and who had normal anal cytology or inflammation with detectable HPV 16/18 (n = 9); and group 6: PLWH who had normal anal cytology or inflammation without HPV 16/18 detection (n = 136). ASCUS, atypical squamous cells of undetermined significance; Gr, group; HPV, human papillomavirus; NMDs, Non-metric Multidimensional Scaling; PcoA: Principal Co-ordinates Analysis; PLWH, persons living with HIV.

In the present study, diminished richness of bacterial diversity was noted in group 1 compared with other groups, i.e., group 2 and group 6. The variability within each group also differed, indicating differences in the spread or dispersion of microbial communities between the groups. Enterobacteriaceae, *Proteus*, Morganellaceae, *Mycoplasma*, *Sutterella*, *Ruminococcus*, and Bacilli were significantly more abundant in group 1 than in groups 2, 3, 4, and 6. In contrast, Bacteroidales and Clostridia were abundant in group 6.

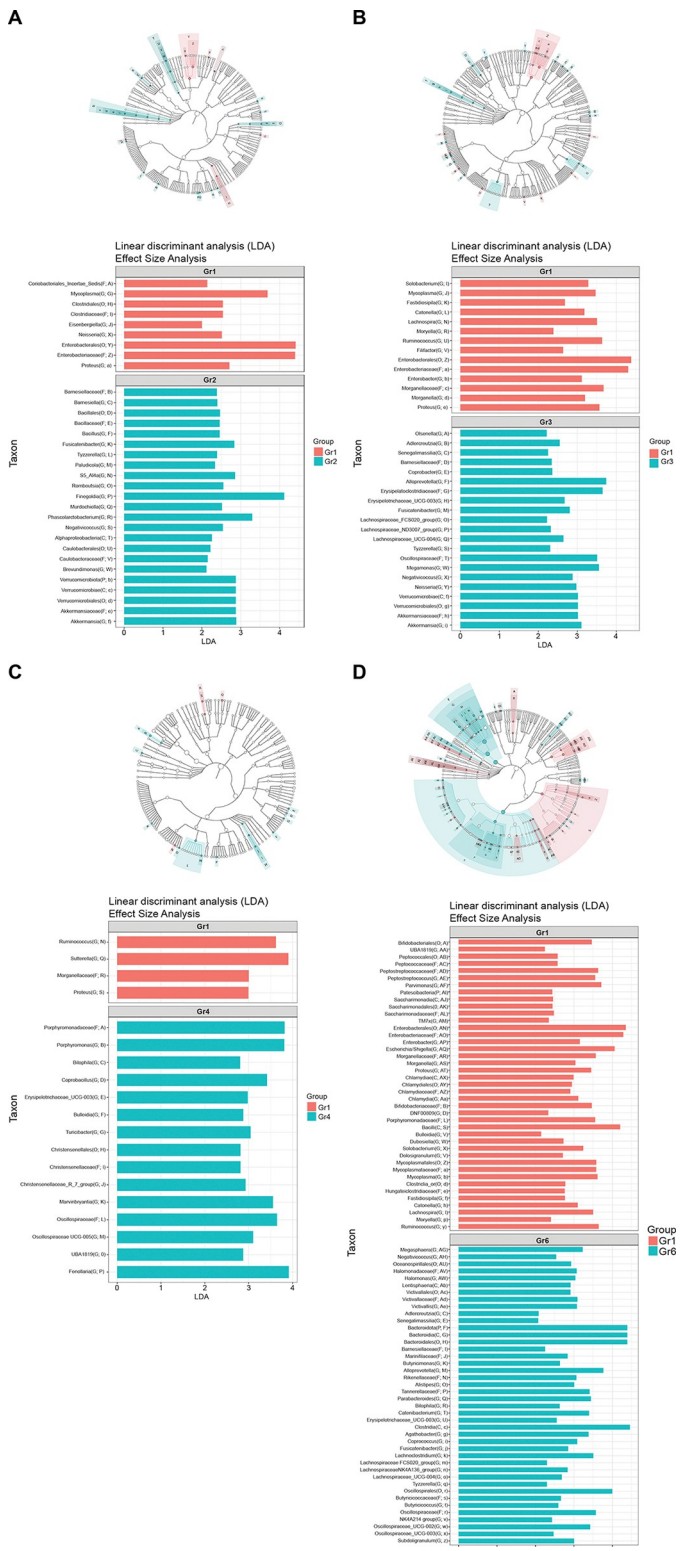

**Fig 4. Linear discriminant analysis effect size comparing group 1 and groups 2, 3, 4, 5, and 6 and cladogram.** 4a: group 1 vs. group 2; 4b: group 1 vs. group 3; 4c: group 1 vs. group 4, 4d: group 1 vs. group 6. Note: group 1: PLWH whose CD4+ T cell counts were < 500 cells/μL with anal cytology yielding ASCUS+ and detectable HPV 16/18 (n = 21); group 2: PLWH whose CD4+ T cell counts were ≥ 500 cells/μL with anal cytology yielding ASCUS+ with detectable HPV 16/18 (n = 25); group 3: PLWH who had anal cytology yielding ASCUS+ without HPV 16/18 detection

(n = 87), group 4: PLWH whose CD4+ T cell counts were < 500 cells/μL and who had normal anal cytology or inflammation with detectable HPV 16/18 (n = 13); group 5: PLWH whose CD4+ T cell counts were ≥ 500 cells/μL and who had normal anal cytology or inflammation with detectable HPV 16/18 (n = 9); and group 6: PLWH who had normal anal cytology or inflammation without HPV 16/18 detection (n = 136). ASCUS, atypical squamous cells of undetermined significance; Gr, group; HPV, human papillomavirus; PLWH, persons living with HIV.

Dinh et al. studied 21 PLWH and found the relative abundance of intestinal Proteobacteria, Gammaproteobacteria, Enterobacterales, and Enterobacteriaceae was significantly enriched compared to HIV-negative controls [45]. Dubourg et al. studied 31 PLWH and showed that PLWH has an increased relative abundance of gut Gammaproteobacteria compared to HIV-negative cases [46]. Previous researchers have conducted comprehensive reviews highlighting increased inflammatory biomarkers, including CD14, IL-6, and CD38+HLA-DR+CD8+ T cells in the gut, as well as increases in traditionally pathogenic bacteria such as Enterobacteriaceae, including Morganellaceae and *Proteus* [46, 47]. Similar to the current study, we also demonstrated a higher relative abundance of Enterobacterales and Enterobacteriaceae in group 1 compared to other groups. However, these previous studies did not examine the role of oncogenic HPVs or the differences in anal cytology findings. *Mycoplasma* is the smallest intracellular bacterium that establishes persistent intracellular infections that can lead to inflammatory cytokine-mediated tissue injury. In addition, *Mycoplasma* infection allows direct interaction with HPV during co-infection of epithelial cells [48], an observation consistent with our findings. *Sutterella* is a gram-negative, non-spore-forming rod grown under anaerobic or microaerophilic conditions. It belongs to the family Betaproteobacteria, is proinflammatory [49], and could explain our findings regarding the abundance of *Sutterella* in the anal samples of group 1 patients. *Ruminococcus*, a gram-positive, spore-forming, anaerobic rod, is a polyphyletic genus with species belonging to two Firmicutes families, Lachnospiraceae and Ruminococcaceae, capable of degrading cellulose. Ron et al. studied 128 MSM, of whom 47 (36.7%) had biopsy-proven HSIL, and found that *Ruminococcus* is a predictor of HSIL [44]. Although the richness of this anerobic commensal, *Ruminococcus*, was found to decrease in the gut samples of PLWH [50], caution should be exercised when it is found in abundance at anal sites, particularly the Ruminococcaceae NK4A214 group [44]. Bacilli have been reported to correlate with the progression of HPV-related cervical lesions [13] and may therefore deserve attention with regard to the progression of anal lesions. Previous studies have mentioned the contribution of *Fusobacterium* and *Sneathia* to HPV-related lesions [29, 43], which could not be proven in this study. Ethnic or environmental factors may contribute to these different findings [13, 29]. The above-mentioned anal bacterial ecology changes with the presence of oncogenic HPVs and the transformation of the anal epithelium.

The strength of this study lies in its interpretation of the anal microbial compositions of a large cohort of MSM-LWH that had different stages of HPV-related disease using high-throughput sequencing. The limitations of this study include its cross-sectional design. Hence, we could not conclude a causal relationship between the anal microbiota and HPV infection or precancerous anal cytology. Second, nearly 100% of the study cohort received HAART; therefore, the findings of this study cannot be generalized to PLWH without HAART. Third, the participants in this study were from a single urban tertiary care hospital, and the results cannot be generalized to different populations. Finally, although in this study we strongly recommended high-resolution anoscopy to obtain the histology results for patients with HPV 16/18 or cytology yielding ASCUS+, less than 20% of the patients actually visited proctologists. The patients' reluctance to undergo anoscopy motivated us to explore non-invasive biomarkers.

The study showed that MSM-LWH with CD4+ T cell counts <500 cells/μL, anal ASCUS+, and detectable HPV16/18 have low bacterial diversity and a relative abundance of *Enterobacteriaceae*, *Ruminococcus*, and Bacilli compared to other participant groups. We concluded that understanding the interaction among immunocompromised hosts, oncogenic HPVs, and microbiota is essential, and the contribution of these factors to anal precancerous lesions needs more in-depth exploration. Furthermore, the anal swab signature microbiota profiles, such as *Lactobacillus* in the cervix, warrant further exploration.

## Supporting information

**S1 Table. Results of the permutational multivariate analysis of variance (adonis function) and the multivariate homogeneity of group dispersions analysis (betadisper function).** (DOCX)

**S1 File. Questionnaire for enrolled participants.** (PDF)

## Acknowledgments

We acknowledge the Core Instrument Center of the Taipei Medical University for providing the next generation sequence service. We also thank the patients and the care team of Comprehensive AIDS Care Center at Taoyuan General Hospital, Ministry of Health and Welfare.

## Author Contributions

**Conceptualization:** Shu-Hsing Cheng, Fang-Yeh Chu, Yun-Ru Liu.

**Data curation:** Shu-Hsing Cheng, Cheng-Pin Chen, Yi-Chun Lin, Chien-Yu Cheng.

**Formal analysis:** Yu-Chen Yang, Kuo-Sheng Liao.

**Funding acquisition:** Shu-Hsing Cheng.

**Investigation:** Shu-Hsing Cheng, Hui-Ting Hsieh.

**Methodology:** Shu-Hsing Cheng, Fang-Yeh Chu, Yun-Ru Liu.

**Project administration:** Shu-Hsing Cheng.

**Resources:** Shu-Hsing Cheng.

**Software:** Yu-Chen Yang.

**Supervision:** Fang-Yeh Chu, Yun-Ru Liu.

**Validation:** Shu-Hsing Cheng, Yu-Chen Yang, Cheng-Pin Chen, Yun-Ru Liu.

**Visualization:** Yu-Chen Yang.

**Writing – original draft:** Shu-Hsing Cheng.

**Writing – review & editing:** Shu-Hsing Cheng, Yu-Chen Yang, Cheng-Pin Chen, Hui-Ting Hsieh, Yi-Chun Lin, Chien-Yu Cheng, Fang-Yeh Chu, Yun-Ru Liu.

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
