## [Decision Letter · Decision Letter 0]

10 Jul 2024

PONE-D-24-17510

Oncogenic human papillomavirus and anal microbiota in men who have sex with men and are living with HIV in Northern Taiwan

PLOS ONE

Dear Dr. Chu,

Thank you for submitting your manuscript to PLOS ONE. After careful consideration, we feel that it has merit but does not fully meet PLOS ONE’s publication criteria as it currently stands. Therefore, we invite you to submit a revised version of the manuscript that addresses the points raised during the review process. Please note that some of the comments (Reviewer 1)  were provided in a separate attachment and the attachment should be accompanying the email as per the note at the end..

Critically, additional clarity in the methods section is necessary and the data should be presented more concisely. Some of the statistical methodology might also be suboptimal.

We look forward to receiving your revised manuscript.

Kind regards,

Ivan Sabol

Academic Editor

PLOS ONE

https://academic.oup.com/jid/article/224/7/1247/6129308?login=false

In your revision ensure you cite all your sources (including your own works), and quote or rephrase any duplicated text outside the methods section. Further consideration is dependent on these concerns being addressed.

Additional Editor Comments:

Please also take a look these more technical issues in addition to the reviewer’s comments:

Page 4 Line 79 listing inclusion and exclusion criteria for invitation to participate explicitly might be helpful. Currently the Line 79 implies MSM was an inclusion criteria but 9 participants failed this (P8L166)?

P6 L123– 124 how many indexes were used in one run? If all 291 samples were assessed by 16rRNA seq this means that batches were made. Was batch effect investigated and how was it addressed?

P6 L127 which flow cell and reagents (cartridge) were used (v2/v3)?

P7 L129 are the raw sequences deposited in a sequence repository?

P7 L138 was some preprocessing / filtering done on the pyloseq object or during preceeding steps? Ie normalization or filtering of taxa?

P7 L140-144 last two paragraphs lack version and or references for the packages used.

P7 L150 testing every combination with t-test is inappropriate without multiple testing correction. Why was Deseq not used since phyloseq is supposed to be analysed with deseq for differential abundance http://joey711.github.io/phyloseq-extensions/DESeq2.html?

P7 L151 6 groups for analysis seems excessive? Possibly this level of detail is appropriate for the supplement tables. It is difficult to confirm but the Figure 1 groups appear to be not mutually exclusive (Groups 3 and 6 appear to have the most participants and those 2 groups do not specify the CD4 cell counts unlike all other groups where the cutoff of 5000 cells was employed.)

P8 L165 it might be more informative if failed cytology and failed HPV exclusions were shown separately

P8 Table 1 “educational year” is worded sub optimally

P8 Table 1 it might be necessary to specify what smoking and drinking represent? Were some guidelines placed in the questionnaire that might be important for interpreting what “yes” answers mean?

P9 Table 1Was the normality of numerical variables tested? Can all reliably be summarized with mean(SD)?

P9 Table 1 were there missing answers in the questionnaires?

P8-9 Table 1 contains excessive white space.

P10 data regarding HPV/cytology and other laboratory measures should be provided in a table for the groups (plus total) discussed instead of text only.

P10 L195 the materials and methods do not adequately introduce the alpha and beta diversity metrics used in the results section. Figure 1 shows some numbers but no context is provided what 500 or 1000 means

P11 L199-201 using t-test for the pairwise comparison of 6 groups is inappropriate. If the data is normally distributed then an ANOVA test with suitable post test should be done.

P11 L201/201 total number of participants in each group should be shown.

Reviewers' comments:

Reviewer's Responses to Questions

**Comments to the Author**

1. Is the manuscript technically sound, and do the data support the conclusions?

Reviewer #1: Yes

Reviewer #2: Yes

2. Has the statistical analysis been performed appropriately and rigorously? 

Reviewer #1: Yes

Reviewer #2: I Don't Know

3. Have the authors made all data underlying the findings in their manuscript fully available?

Reviewer #1: Yes

Reviewer #2: Yes

4. Is the manuscript presented in an intelligible fashion and written in standard English?

Reviewer #1: Yes

Reviewer #2: Yes

5. Review Comments to the Author

Reviewer #1: The manuscript by Chu et al describes the results of a study that performed 16S analysis in anal samples collected from MSM-LWH, characterized in terms of cytology and HPV infection. This is a quite underexplored field of research. The study seems to be well designed and well conducted, although histology is missing and only cytology results are available.

Reviewer #2: In this study “Oncogenic human papillomavirus and anal microbiota in men who have sex with men and are living with HIV in Northern Taiwan”, HPV DNA tests and citology results in anal samples collected Jun 2018 and Dec 2021 in 291 persons living with HIV (PLWH), mainly MSM were analyzed and 16S sequencing was performed to characterize the anal microbiome. Data on the interaction of HPV, HIV and resident commensal bacteria are needed and may be useful in determining the relative risk of lesion progression to anal cancer precursors. Despite the merits of the study, a clearer presentation of the results, additional information and more in-depth discussion are needed in a revised version.

Specific points

Method:

A paragraph on the Bioinformatic analysis methods should be added to explain the tests that are listed in Results: e.g. UniFrac, PcoA on GuniFrac, Non-metric Multidimensional Scaling (NMDs), Bray-Curtis distance (VST), Adonis tests and the Betadisper test.

The paragraph HPV genotyping should be renamed to HPV detection and it should be more clearly stated that individual results for HPV 16 and HPV 18 are given, along with a simultaneous, pooled result for the other oncogenic HPVs.

Was high-resolution anoscopy (HRA) performed after an ASCUS+ result?

Results

The presentation of results should be improved. For example, the distribution of samples in the six groups could be presented in a table.

The individual results for HPV 16, HPV 18 and other oncogenic HPVs should be reported in a table or text and whether positivity was statistically associated with ASCUS+ results. Moreover, how many patients were diagnosed with anal SIL should be reported.

Discussion:

Lines 281-282: “The interaction between microbiota, HPV acquisition and persistence, and the development of cervical cancer in women has been researched in detail.” I would suggest to modify this sentence in: the vaginal microbiota has been studied in detail, but the interactions with HPV acquisition and persistence and cervical cancer risk are still not fully understood. Accordingly, I would suggest to shorten the description of the vaginal microbiota community state types (CSTs) that are well known (lines 283-295) and to focus on studies (if any) about the anal mucosa microbiota in women.

Lines 306-308: “In the pathophysiology of infectious diseases, pathogenic colonization and infection involve dynamic interactions between microbes and the microbiome, host, and environmental factors [39]. This biology is also applicable to PLWH who are MSM” The sentences “This biology is also applicable to PLWH who are MSM” should be rephrased, e.g. in This biology also applies to the anal mucosal environment of PLWH who are MSM.

At line 322: “homogeneity of the dispersion also differed between the groups...” This should be rephrased for a better comprehension of this concept.

At lines 323-325: “Enterobacteriaceae, Proteus, Morganellaceae, Mycoplasma, Sutterella, Ruminococcus, and Bacilli were significantly abundant in PLWH whose CD4+ T cell counts were < 500 cells/µL and who had anal cytology yielding ASCUS or higher grades with detectable HPV 16/18 (group 1).” As this sentence implies a comparison, other group(s) should be mentioned.

At lines 328-331: “HIV infection also frequently correlates with increased tissue and circulating measures of inflammation, such as CD14, IL-6, and CD38+HLA-DR+CD8+ T cells in the gut [41], as well as increases in traditionally pathogenic bacteria such as Enterobacteriaceae, including Morganellaceae and Proteus [42,43].” The sentence should be reworded to more clearly relate the results of the references mentioned here [41-43] to the results of this study.

The same applies to the sentences at lines 342-343: “Our findings are consistent with those of Ron et al., who described Ruminococcus-predicting HSIL [40]. In contrast, a decreased abundance of Ruminococcus was previously noted in the guts of PLWH [46].”

and to the sentences at lines 353-354 (limitations): “Third, previous studies have mentioned the contribution of Fusobacterium and Sneathia to HPV-related lesions [22,39], which could not be proven in this study” Ref 39 (Pathogens, microbiome and the host: emergence of the ecological Koch's postulates) is a review; hence, the specific study reporting about Fusobacterium and/or Sneathia should be specified and discussed.

6. PLOS authors have the option to publish the peer review history of their article (what does this mean?). If published, this will include your full peer review and any attached files.

Reviewer #1: No

Reviewer #2: No

---

## [Author Response · Author response to Decision Letter 0]

15 Aug 2024

Response to reviewers uploaded at Attach Files step.

---

## [Decision Letter · Decision Letter 1]

17 Sep 2024

PONE-D-24-17510R1Oncogenic human papillomavirus and anal microbiota in men who have sex with men and are living with HIV in Northern TaiwanPLOS ONE

Dear Dr. Chu,

Thank you for submitting your manuscript to PLOS ONE. After careful consideration, we feel that it has merit but does not fully meet PLOS ONE’s publication criteria as it currently stands. Therefore, we invite you to submit a revised version of the manuscript that addresses the points raised during the review process. The reviewer comments were sufficiently addressed. However, PLOS One publication criteria emphasize the need for high standard and rigor in data analysis and statistics.Therefore the problems with potentially inappropriately performed statistics and uncertain p values derived from this analysis must be addressed.

We look forward to receiving your revised manuscript.

Kind regards,

Ivan Sabol

Academic Editor

PLOS ONE

Journal Requirements:

Additional Editor Comments:

The material and methods revisions are appreciated and improved the clarity of the manuscript

Some small methodological problems remain

P7 Line 143-144 grammatically incorrect and possibly misplaced. Batch effect would be compensated after trimming and denoising.

Page 8 Line 173 statistical analysis section

The authors reply “Line 170: We used the Wilcoxon test, considering it as a better alternative to the ttest when the normal distribution of the differences between paired individuals cannot be assumed.“

However, there are several issues with this revision.

1)The revision deleted the mention of t-test from Statistical analysis section at line 177 leaving only chi-square test mentioned

2) The Figure 2 p values, the Figure 2 legend, or the text at P12 L227 remains unchanged suggesting that it still shows the old t-test individual p values. The test used to obtain the p values is undeclared and inconsistent with the paragraph Statistical analysis

3) The comparisons between the six groups presented at Figure 2 cannot be considered paired individuals and Wilcoxon test cannot be considered as appropriate for these comparisons

P12 Table 2 footer would benefit from listing the 12 other HPV types as a footnote

Supplementary table 1 was uploaded twice. (once as Table_S1.docx and once as S1_table.docx) on pages 40 and 41 of the combined PDF

Reviewers' comments:

Reviewer's Responses to Questions

**Comments to the Author**

1. If the authors have adequately addressed your comments raised in a previous round of review and you feel that this manuscript is now acceptable for publication, you may indicate that here to bypass the “Comments to the Author” section, enter your conflict of interest statement in the “Confidential to Editor” section, and submit your "Accept" recommendation.

Reviewer #2: All comments have been addressed

2. Is the manuscript technically sound, and do the data support the conclusions?

Reviewer #2: Yes

3. Has the statistical analysis been performed appropriately and rigorously? 

Reviewer #2: Yes

4. Have the authors made all data underlying the findings in their manuscript fully available?

Reviewer #2: Yes

5. Is the manuscript presented in an intelligible fashion and written in standard English?

Reviewer #2: Yes

6. Review Comments to the Author

Reviewer #2: (No Response)

7. PLOS authors have the option to publish the peer review history of their article (what does this mean?). If published, this will include your full peer review and any attached files.

Reviewer #2: No

---

## [Author Response · Author response to Decision Letter 1]

3 Oct 2024

Response to reviewers

Comment: Please review your reference list to ensure that it is complete and correct. If you have cited papers that have been retracted, please include the rationale for doing so in the manuscript text, or remove these references and replace them with relevant current references. Any changes to the reference list should be mentioned in the rebuttal letter that accompanies your revised manuscript. If you need to cite a retracted article, indicate the article’s retracted status in the References list and also include a citation and full reference for the retraction notice.

Reply: We have reviewed and checked the reference list.

Comment: P7 Line 143-144 grammatically incorrect and possibly misplaced. Batch effect would be compensated after trimming and denoising.

Reply: We have corrected the grammar and revised the sentence to “Batch effects were compensated after trimming and denoising” and moved it to an appropriate location on page 7, line 150.

Comment: Page 8 Line 173 statistical analysis section

The authors reply “Line 170: We used the Wilcoxon test, considering it as a better alternative to the ttest when the normal distribution of the differences between paired individuals cannot be assumed.“

However, there are several issues with this revision.

1)The revision deleted the mention of t-test from Statistical analysis section at line 177 leaving only chi-square test mentioned

2) The Figure 2 p values, the Figure 2 legend, or the text at P12 L227 remains unchanged suggesting that it still shows the old t-test individual p values. The test used to obtain the p values is undeclared and inconsistent with the paragraph Statistical analysis

3) The comparisons between the six groups presented at Figure 2 cannot be considered paired individuals and Wilcoxon test cannot be considered as appropriate for these comparisons.

Reply: We apologize for the confusion regarding the statistical methods. To clarify:

1) The p-values in the figures were derived from Wilcoxon tests (the default method in the analysis package). Student t-tests were never used in this study. We have now explicitly mentioned the use of the Wilcoxon–Mann–Whitney test at both line 173 (in the "16s rRNA gene sequence analysis" section) and line 170 of the Statistical analysis section.

2) We have corrected Figure 2, its legend, and the text at P12 L227 to reflect the use of Wilcoxon tests. The statistical methods are now explicitly stated in the Figure 2 legend and at P13 L243.

3) In Figure 2, we have shown a comparison of group 1 (PLWH with CD4+ T cell counts < 500 cells/µL, ASCUS+, and detectable HPV 16/18) with each of the other groups individually using the Wilcoxon test, rather than comparing all six groups simultaneously. The Kruskal–Wallis test was not used.

Comment: P12 Table 2 footer would benefit from listing the 12 other HPV types as a footnote

Reply: We have added the 12 additional HPV types as a footnote in Table 2.

Comment: Supplementary table 1 was uploaded twice. (once as Table_S1.docx and once as S1_table.docx) on pages 40 and 41 of the combined PDF

Reply: Thank you for pointing this out. We have corrected the duplication of Supplementary Table 1.

---

## [Editor Report · Decision Letter 2]

8 Oct 2024

Oncogenic human papillomavirus and anal microbiota in men who have sex with men and are living with HIV in Northern Taiwan

PONE-D-24-17510R2

Dear Dr. Chu,

We’re pleased to inform you that your manuscript has been judged scientifically suitable for publication and will be formally accepted for publication once it meets all outstanding technical requirements.

Kind regards,

Ivan Sabol

Academic Editor

PLOS ONE